# PROCEEDINGS A

## Perspective

atmospheric chemistry, environmental chemistry, oceanography

iodine, iodide, halogens, sea–air interactions, ozone, global iodine cycle

**Author for correspondence:**
Lucy J. Carpenter
e-mail: lucy.carpenter@york.ac.uk

†Present address: Ricardo Energy and Environment, Harwell, Oxfordshire, UK.
‡Present address: IMT Lille Douai, University of Lille, CERI EE - Sciences de L'Atmosphère et Génie de L'Environnement, 59000 Lille, France.

An invited perspective to mark the election of Lucy Carpenter as a Fellow of the Royal Society in 2019.

# Marine iodine emissions in a changing world

Lucy J. Carpenter[1], Rosie J. Chance[1],
Tomás Sherwen[1,2], Thomas J. Adams[3,†],
Stephen M. Ball[3], Mat J. Evans[1,2], Helmke Hepach[4],
Lloyd D. J. Hollis[3], Claire Hughes[4],
Timothy D. Jickells[5], Anoop Mahajan[7],
David P. Stevens[6], Liselotte Tinel[1,‡] and
Martin R. Wadley[6]

[1]Wolfson Atmospheric Chemistry Laboratories, Department of Chemistry, University of York, York, UK
[2]National Centre for Atmospheric Science (NCAS), University of York, York YO10 5DD, UK
[3]School of Chemistry, University of Leicester, Leicester, UK
[4]Department of Environment and Geography, University of York, Wentworth Way, Heslington, York, UK
[5]Centre for Ocean and Atmospheric Sciences, School of Environmental Sciences, and [6]Centre for Ocean and Atmospheric Sciences, School of Mathematics, University of East Anglia, Norwich Research Park, Norwich, UK
[7]Indian Institute of Tropical Meteorology, Ministry of Earth Sciences, Pune 411008, India

LJC, 0000-0002-6257-3950; RJC, 0000-0002-5906-176X;
TS, 0000-0002-3006-3876; SMB, 0000-0002-5756-4718;
MJE, 0000-0003-4775-032X; LT, 0000-0003-1742-2755

Iodine is a critical trace element involved in many diverse and important processes in the Earth system. The importance of iodine for human health has been known for over a century, with low iodine in the diet being linked to goitre, cretinism and neonatal death. Research over the last few decades has shown that iodine has significant impacts on tropospheric photochemistry, ultimately impacting climate by reducing the radiative forcing of ozone ($O_3$) and air quality by reducing extreme $O_3$

concentrations in polluted regions. Iodine is naturally present in the ocean, predominantly as aqueous iodide and iodate. The rapid reaction of sea-surface iodide with $O_3$ is believed to be the largest single source of gaseous iodine to the atmosphere. Due to increased anthropogenic $O_3$, this release of iodine is believed to have increased dramatically over the twentieth century, by as much as a factor of 3. Uncertainties in the marine iodine distribution and global cycle are, however, major constraints in the effective prediction of how the emissions of iodine and its biogeochemical cycle may change in the future or have changed in the past. Here, we present a synthesis of recent results by our team and others which bring a fresh perspective to understanding the global iodine biogeochemical cycle. In particular, we suggest that future climate-induced oceanographic changes could result in a significant change in aqueous iodide concentrations in the surface ocean, with implications for atmospheric air quality and climate.

## 1. Iodine in the atmosphere

Atmospheric iodine is mainly derived from the oceans, which contain approximately 70% of the Earth's surface inventory of natural iodine [1]. Volatilization of oceanic iodine, and in smaller amounts, terrestrial iodine, to the atmosphere involves both biological and non-biological pathways [2–5]. The ease of volatilization of iodine, in both inorganic and organic forms, is considerably greater than that of chlorine and bromine and this aspect of its geochemistry makes iodine unique among the halogens.

The role of iodine on the atmosphere was first explored by Chameides & Davis [6], who used a photochemical model to infer significant impacts on tropospheric photochemistry caused predominantly by oceanic emissions of methyl iodide ($CH_3I$). Observational evidence of the widespread impacts of reactive iodine came nearly three decades later [7,8], confirming that iodine could be highly significant in influencing atmospheric photochemistry over the oceans. Since then, several observational studies have confirmed the ubiquitous presence of iodine oxide radicals (IO) in the marine troposphere [9–15].

As indicated in figure 1, gaseous iodine compounds emitted from the ocean are rapidly (minutes to days) photolysed in the atmosphere to produce iodine atoms, which react with $O_3$ in the atmosphere to form the IO radical. IO can be considered a 'smoking gun' for the presence of active iodine chemistry. It reacts further with nitrogen and hydrogen oxides to perturb important aspects of atmospheric chemistry. Due to its significant role in a multitude of atmospheric processes, atmospheric iodine cycling is now being incorporated into chemical transport and air quality models. These models show that iodine has a profound impact on tropospheric photochemistry, causing a reduction in tropospheric $O_3$ (a key climate and air quality gas) of approximately 15% globally [18,19], reducing summertime $O_3$ exposure over Europe by around approximately 15% [20], representing an important negative feedback mechanism on $O_3$ [21–23] and acting as a source of aerosols [19,24,25]. In addition, iodine has recently been shown to be injected into the stratosphere, where it may represent a small but significant contribution to $O_3$ depletion [26,27]. However, critical uncertainties remain in determining the impacts of iodine on the atmosphere and how they may change in the future.

Over the last decade or so, evidence has emerged that oceanic emissions of iodinated organic compounds such as $CH_3I$, $CH_2ICl$ and $CH_2I_2$ are likely not the primary source of atmospheric iodine, as was originally thought, but may comprise only around 20% of the total iodine flux to the atmosphere globally [13,21,28]. The dominant fraction (80%) is instead believed to arise from the heterogeneous reaction of iodide ($I^-_{(aq)}$) with gaseous $O_3$ at the sea surface, releasing $I_2$ and HOI [17,21,29] (reactions 1–6). However $CH_3I$, as one of the longer-lived precursors, constitutes an important source of iodine above the marine boundary layer [26]. Emission inventories for the iodocarbons have been compiled by Bell *et al.* [30] and Ordóñez *et al.* [31].

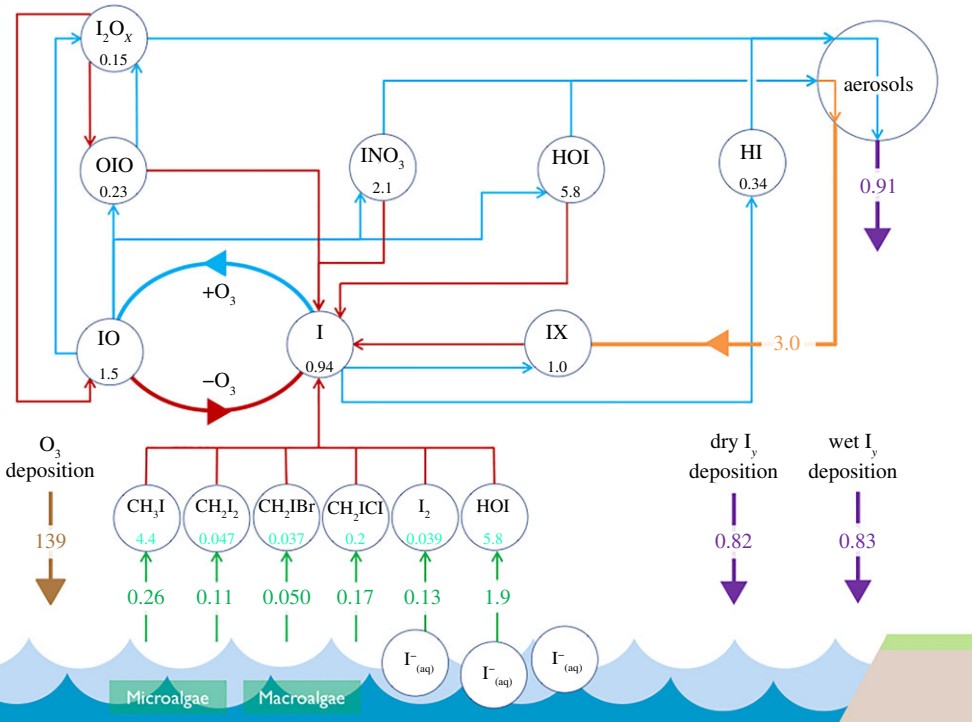

**Figure 1.** Schematic to illustrate tropospheric iodine chemistry. Average global annual mean burdens (Gg I) are shown below key $I_Y$ species (I, $I_2$, HOI, IO, OIO, HI, INO, $INO_2$, $INO_3$, $I_2O_x$, $I_2O_3$, $I_2O_4$), with global total fluxes shown on arrows (Tg(I) year$^{-1}$). Red lines, photolysis; blue lines, chemical pathways; green lines, emission source; orange lines, heterogeneous pathway; purple lines, depositional pathway. Ozone deposition to the oceans (Tg($O_3$) year$^{-1}$) is also shown in brown to illustrate the driving force behind the inorganic emissions. Adapted with permission from [16] and updated to give values from the GEOS-Chem (v. 12.9.1) model, which uses sea-surface iodide fields from [17]. (Online version in colour.)

$$I^-_{(aq)} + O_{3(g \text{ or } aq)} \rightarrow IOOO^-, \tag{R1}$$

$$IOOO^- \rightarrow IO^-_{(aq)} + O_2, \tag{R2}$$

$$IO^-_{(aq)} + H^+ \rightarrow HOI_{(aq)}, \tag{R3}$$

$$HOI_{(aq)} + I^-_{(aq)} + H^+ \rightleftharpoons I_{2(aq)} + H_2O, \tag{R4}$$

$$I_{2(aq)} \rightleftharpoons I_{2(g)}, \tag{R5}$$

and $$HOI_{(aq)} \rightleftharpoons HOI_{(g)}. \tag{R6}$$

The estimated magnitude of global sea–air emissions of $I_2$ and HOI, as shown in figure 1, originate from the laboratory work and parametrizations of Carpenter *et al.* [21] and MacDonald *et al.* [17], mostly using experiments with iodide solutions. We have recently confirmed these parametrizations for $I_2$ emissions from ozone reactions with iodide-containing solutions, using a sensitive broadband cavity-enhanced absorption spectroscopy (BBCEAS) method to measure at ambient conditions [29]. All these studies have noted that $I_2$ emissions from real seawater are lower than from artificial iodide solutions, most likely due to the presence of (poorly characterized) oceanic dissolved organic material and/or surfactants (see review by Hansell & Carlson [32]). There were insufficient data in the original studies [17,21] to parametrize the reduction in $I_2$ emissions or to identify the mechanism/s involved. Several studies have found that added organic material, depending on its properties, can either reduce or increase $I_2$ emissions from aqueous solution, due to different physical or chemical mechanisms [33–36].

In the first study using samples of the sea-surface microlayer (SML), we have confirmed that $I_2$ emissions are decreased compared to artificial seawater by up to a factor of 10, and suggest that this reduction likely arises from the increased solubility of $I_2$ in the organic-rich interfacial layer of seawater [29]. However, this study was carried out using only limited SML samples from the same location in the North Sea. More data are required to confirm how the SML in other locations affects $I_2$ emissions and, importantly, to determine any influence on HOI, which is the single largest contributor to iodine emissions (figure 1).

Global modelling studies [18,37] have shown that the existing parametrizations for oceanic inorganic iodine ($I_2$ and HOI) release, alongside climatologies of organic iodine emissions [31], result in reasonable simulation of the tropospheric IO observations which have been made in different oceanic regions [7,10,13,38]. However, a recent study with simultaneous observations of IO, $O_3$ and sea-surface iodide in/over the Indian Ocean and the Southern Ocean found that the observed IO concentrations could not be adequately computed using inorganic iodine fluxes and the currently understood chemistry [15]. Calculated sea–air fluxes of HOI and $I_2$, using the MacDonald *et al.* [17] global iodide fields and measured $O_3$ concentrations, were used as inputs to two independent global atmospheric chemistry models (GEOS-Chem and CAM-Chem). Both models suggested higher than observed IO levels in the Indian Ocean region but under-predicted [IO] for the Southern Ocean region [15]. However, although there was also no correlation between measured and modelled IO levels across the entire dataset, the GEOS-Chem modelled IO showed a significant positive correlation with observed IO above the 99% confidence limit for data north of the polar front. These discrepancies highlight the major uncertainties which still exist in our understanding of iodine biogeochemistry and call for further studies of IO and related halogenated species in the ocean and atmosphere. There is also a need for further studies relevant to polar regions, where elevated levels of IO [39] and iodine associated with new particle formation events [24,40] have been detected. There is evidence of release of iodine (and other halogens) from sea-ice (e.g. [41] and references therein) but, as yet, no consensus on the contribution of various iodine sources to the polar atmosphere.

Alongside its contribution to atmospheric iodine emissions, sea-surface iodide has also been identified as an important depositional sink for tropospheric ozone [42,43]. Dry deposition of $O_3$ to the Earth's surface is estimated to account for about 25% of overall tropospheric $O_3$ removal. Loss to the ocean surface, via reaction R1, is believed to represent the largest single depositional sink by land cover class [44,45]; this impact of iodine on $O_3$ is in addition to the gas phase catalytic cycles occurring in the atmosphere. Model calculations show that reaction R1 has the potential to reduce surface $O_3$ mixing ratios through $O_3$ deposition alone by several ppb [46–48], which is of a magnitude where it can influence human exposure and impact on ecosystems and agricultural crop yields. However, both the mechanistic details and the rates of oceanic $O_3$ deposition are subject to much greater uncertainty than deposition to land, which translates into large differences in the predicted global ocean dry deposition flux [44].

## 2. Sea-surface iodide

Sea-surface iodide, which can vary from low nanomolar concentrations at high latitudes to several 100 nM in tropical and coastal seas (table 1) [49], is a critical factor in controlling both atmospheric iodine concentrations and oceanic dry deposition of $O_3$, via reaction (R1). Atmospheric models have used parametrizations of iodide concentrations to provide boundary conditions for global iodide fields [18,46,47,53]. Such parametrizations have generally fitted relatively small numbers of sea-surface iodide observations to simple functions using proxies for iodide such as dissolved nitrate and sea-surface temperature [17,43,49]. We recently updated our global compilation of iodide observations (1967–2018) [50], resulting in a 45% larger sample size ($n = 1342$) than described previously ($n = 925$; table 1, [49]). The new data include a large number of new observations from the previously very under-sampled Indian Ocean basin [54], so large-scale sea-surface iodide transects are now available for all ocean basins except the Arctic [50]. Using the expanded dataset and a machine learning random forest regression (RFR) approach, we have

**Table 1.** Summary statistics for observed and predicted global sea-surface iodide fields. Predicted values are annually averaged. Model simulations are for the present day. Note differences in maximum predicted values may arise from differences in the way very high observational data points have been treated when developing models.

| | [Iodide], nM | | | | | |
|---|---|---|---|---|---|---|
| | mean | standard deviation | lower quartile | median | upper quartile | maximum |
| *observational datasets* | | | | | | |
| Chance *et al.* [49] ($n = 925$) | 92 | 81 | 28 | 77 | 140 | 700 |
| Chance *et al.* [50] ($n = 1342$) | 108 | 111 | 38 | 89 | 147 | 2039 |
| *sea-surface temperature parametrizations* | | | | | | |
| MacDonald *et al.* [17] | 59 | 35 | 17 | 51 | 87 | 126 |
| Chance *et al.* [49] | 128 | 65 | 49 | 122 | 179 | 226 |
| *multivariate machine learning parametrization* | | | | | | |
| Sherwen *et al.* [51] | 106 | 46 | 52 | 106 | 139 | 220 |
| *global biogeochemical model simulation* | | | | | | |
| Wadley *et al.* [52] | 122 | 75 | 65 | 100 | 149 | 973 |

generated a high-resolution (0.125° × 0.125°, ~12.5 km × 12.5 km) monthly dataset of present-day global sea-surface iodide [51]. The iodide observations were used as the dependent variable and co-located ancillary parameters (including temperature, mixed layer depth, salinity and nitrate) from global climatologies as the independent variables.

As shown in figure 2*a*, the predicted iodide distributions from the previous statistical relationships [17,49] and from Sherwen *et al.* [51] all reflect the large-scale observed ocean distribution of iodide, with high concentrations in low latitude warm waters, and low iodide concentrations at high latitudes in seasonally overturning cold waters [49,50]. The iodide concentrations calculated in Sherwen *et al.* [51], using the RFR approach, better capture the observed spatial variability and produce significantly higher concentrations (40% on a global basis) than the commonly used MacDonald *et al.* [17] parametrization (figure 2*b*).

However, while these statistical relationships provide a generally good fit to the observational dataset, their extrapolation beyond the data range for which the relationships are derived cannot be carried out with confidence, nor do they allow prediction of whether oceanic surface iodide may change in the future, or how it may have changed in the past. A detailed process-based knowledge of ocean iodine cycling and its feedbacks with changing environmental parameters is required for such predictive capability.

Below, we describe the current state of knowledge regarding iodine cycling in the ocean, and our recent work to develop a prototype ocean iodine model as a first step towards predicting global oceanic iodine distributions.

## 3. Iodine in the ocean

In the ocean, aqueous iodide ($I^-$, reduced form) and iodate ($IO_3^-$, oxidized form) are the dominant iodine species, with a total concentration of generally 400–500 nM [49]. Thermodynamically, iodate is the favoured form of iodine (except in very oxygen-depleted waters) and it is the overwhelmingly dominant form below the oceanic mixed layer in oxygenated seawater. The presence and distribution of iodide in the surface ocean is essentially determined by its biologically mediated interconversion with iodate and the processes of physical mixing and advection [49,52,56,57]. In the euphotic zone, reduction of iodate to iodide, which has been linked to primary productivity (e.g. [58–60]), means that up to 50% of iodine may be found as iodide. The iodate to iodide transformation has been observed to take place in natural seawater on a timescale

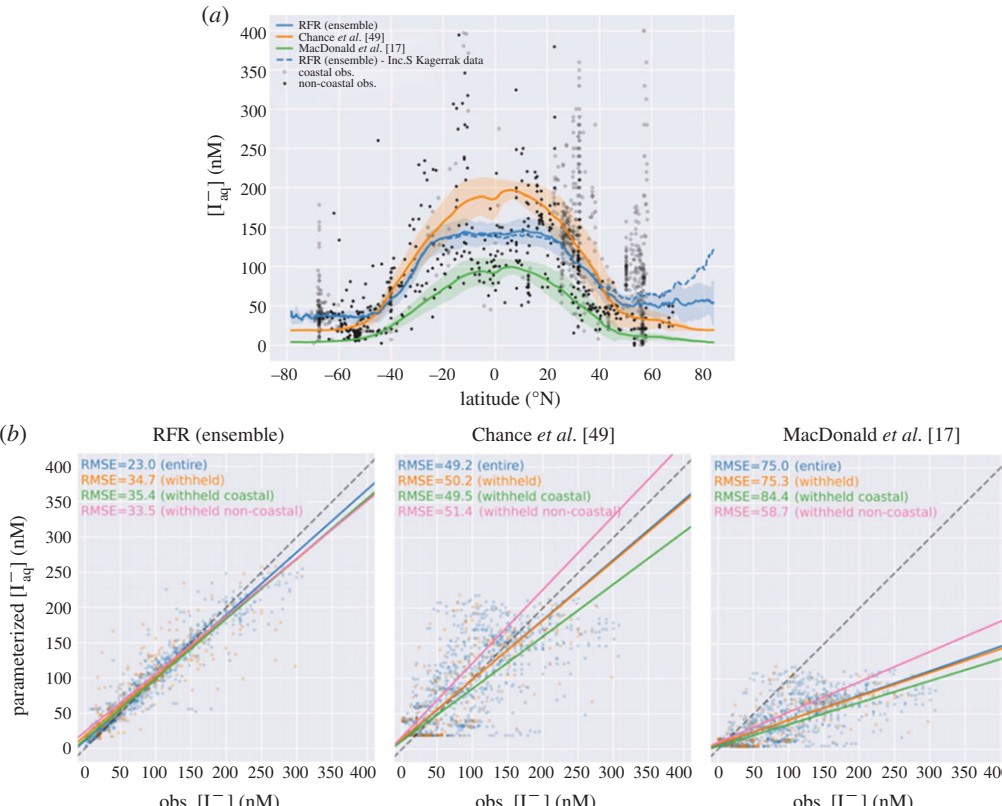

**Figure 2.** (*a*) Predicted latitudinal average sea-surface iodide plotted against latitude, overlaid with observed concentrations, from [51]. Solid lines give the mean values and shaded regions give ± the average standard deviation. For the random forest regression (RFR) ensemble, the standard deviation is the monthly standard deviation within all ensemble members. Black filled diamonds show non-coastal observations and unfilled ones show coastal values. The blue dashed line shows the prediction including data from the Skagerrak strait [55]. The latitudinal range of the horizontal axis is limited to latitudes not permanently covered by sea-ice or land. (*b*) Regression plots showing comparisons between predicted values and observations in the entire dataset (blue, $n = 1293$) and 'withheld' data not used in the prediction (orange, $n = 259$), 'withheld' data classed as coastal (green, $n = 157$) and the 'withheld' data classed as non-coastal (pink, $n = 102$). Solid lines give the orthogonal distance regression line of best fit. The dashed grey line gives the 1 : 1 line. Root mean square error (RMSE) for each line is annotated in each subplot in nanomolar (nM). (Online version in colour.)

of days to weeks [61], but the mechanism is poorly understood, and it is not known whether it takes place by an assimilatory process or as an extracellular or cell surface/dissimilatory reaction. Reduction by nitrate reductase enzymes [62] and reactions of iodate with reduced sulfur species released from cells during senescence [59] have been proposed, but neither of these mechanisms has been confirmed as a significant route of conversion.

Once formed, kinetically stable iodide oxidizes back to iodate slowly; however, this process is highly uncertain with estimated lifetimes ranging from approximately six months to approximately 40 years [49,63]. Earlier estimates of iodide oxidation rate have typically relied upon mass balance approaches (e.g. [56]). Recent measurements made using a radiotracer approach have yielded oxidation rates for natural seawater consistent with these estimates (118–189 nM yr$^{-1}$; [63]). Except for processes specific to the sea-surface microlayer, such as oxidation by $O_3$, rates of chemical oxidation of iodide to iodate in seawater are too slow to account for the observed distribution of iodine species [64], and the process has been assumed to be biologically mediated. The uncertainty around the rates and processes involved has been suggested to be

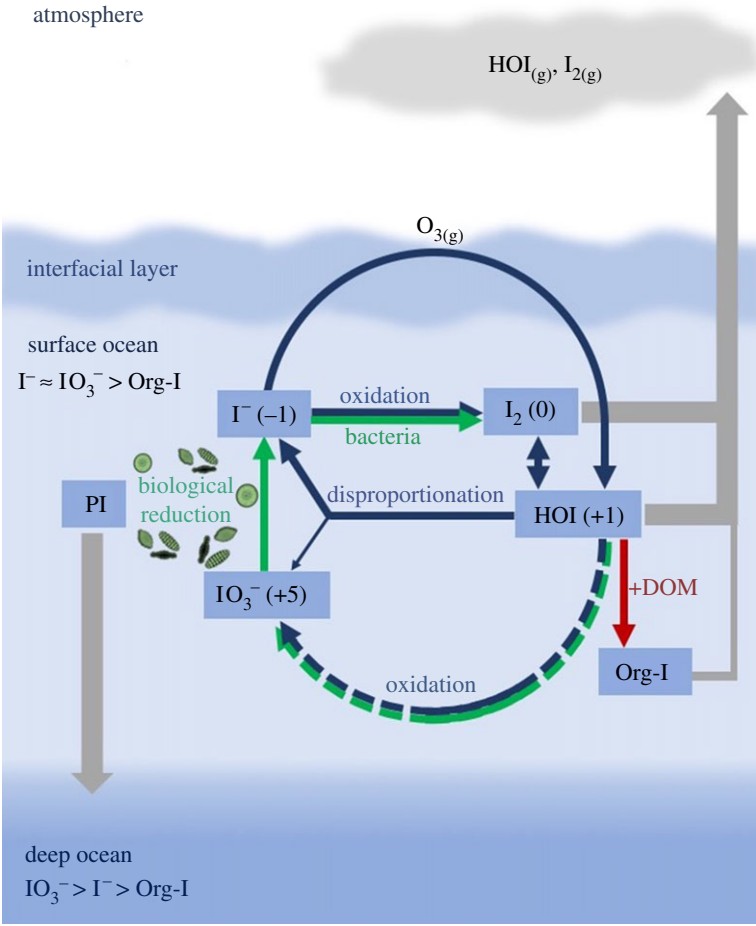

**Figure 3.** Simplified schematic of iodine cycling in the surface ocean. Green lines represent biologically mediated reactions and blue lines are abiotic reactions. 'Org-I' represents dissolved organic iodine, which is observed in coastal surface waters but appears to be a very minor component of total iodine in open ocean waters (see [52] and references therein). (Online version in colour.)

a major limitation to the development of ocean models of iodine transformations [65]. Based on water column observations, it has been proposed that the oxidation of iodide to iodate may be associated with bacterial nitrification [65,66]. Our work has recently confirmed this for laboratory cultures of ammonium-oxidizing bacteria. We observed a significant increase in iodate concentrations compared to media-only controls in the ammonium-oxidizing bacteria *Nitrosomonas* sp. (Nm51) and *Nitrosoccocus oceani* (Nc10) supplied with $I^-$ and $NH_4^+$, indicating that iodide oxidation to $IO_3^-$ is linked to nitrification, and specifically ammonium oxidation [67]. Cell-normalized production rates were 15.69 ($\pm$4.71) fmol $IO_3^-$ cell$^{-1}$ d$^{-1}$ for *Nitrosomonas* sp. and 14.35 ($\pm$8.35) fmol $IO_3^-$ cell$^{-1}$ d$^{-1}$ for *Nitrosococcus oceani*. Nitrification is known to occur throughout the oceanic water column [68], suggesting that iodide oxidation to iodate could be widespread throughout the world's oceans. This mechanism provides an alternative or complementary linkage of the ocean iodine and nitrogen cycles, as suggested by others [69].

Figure 3 summarizes the major components of the upper ocean iodine cycle in oxygenated waters and its interaction with the atmosphere.

The above summarizes the key biogeochemical transformations of iodine species in oxygenated waters. In low oxygen (less than approx. 10 µmol kg$^{-1}$) environments, such as oxygen

minimum zones (OMZs) associated with coastal upwelling systems and anoxic basins such as the Black Sea, iodide is thermodynamically stable and tends to dominate aqueous iodine speciation (e.g. [70–72]). However, evidence from the Pacific boundary systems suggests that iodine speciation in such systems is not a simple function of oxygen concentration and that substantial iodate concentrations may sometimes persist due to slow and variable reduction rates [63,73]. Furthermore, very high 'excess' iodide concentrations of up to several micromolar have been observed in association with low oxygen waters [54,72–74]. Such concentrations cannot be accounted for by the reduction of iodate in the water column, and instead have been ascribed to the release of iodide from sediments under low oxygen conditions [54,72,74]. While low oxygen waters are typically subsurface, elevated iodide associated with such systems has been observed to outcrop at the ocean surface [54,55,72,73] and hence can affect ozone deposition and iodine emission from the sea surface at a local scale. As the extent of ocean deoxygenation increases (e.g. [75]), the incidences of elevated iodide at the sea surface could become more frequent in the future, with possible impacts on atmospheric chemistry.

## 4. Development of an ocean iodine cycling model

Based on the current knowledge of oceanic iodine processes, as discussed above, we have developed the first ocean iodine biogeochemical cycling model that incorporates surface ocean iodine cycling and circulation [52]. Our model captures the processes responsible for determining surface iodine speciation on seasonal timescales, and the accumulated impact of this over the timescale of the circulation in the upper ocean. We do not include any process-based iodine transformations in the deep ocean, but assume a constant iodate concentration, as observed in the current ocean. An iodine cycle has also been incorporated into the cGenie Earth System Model by Lu *et al*. [76], but this model is concerned with very much longer geological timescales, and in particular estimating the particulate I:Ca ratio as a tool for reconstructing trends in upper ocean oxygenation. It represents the surface transformations much more simply, but includes processes in the deep ocean which are likely to change iodine speciation and hence the sedimentary I:Ca ratio on geological timescales.

Our model comprises a three-layer advective and diffusive ocean circulation model of the upper ocean based on the OCCAM ocean GCM and an iodine cycling model embedded within this circulation which allows transformations between the two primary reservoirs of iodine, $IO_3^-$ and $I^-$, allowing prediction of upper ocean iodine speciation. Because of the relatively long lifetimes of iodine species in seawater, advection and mixing have a strong influence on their spatial distributions, and a coupled ocean circulation-biogeochemical model is essential to describe ocean iodine cycling.

Iodide production (from iodate reduction) only occurs in the mixed layer in the model and is driven by monthly averaged primary productivity, linked by an iodine to carbon (I:C) ratio consistent with the values reported in the literature [60,61,77,78]. A satisfactory model fit with observations cannot be obtained with a globally constant I:C ratio, but the best fit is obtained when the I:C ratio is dependent on sea-surface temperature, increasing by an order of magnitude between low and high latitudes. A variation in I:C ratio with sea-surface temperature could be due to different types of plankton dominating primary production under different oceanographic conditions. We assume that the biologically mediated conversion of iodate to iodide occurs during the senescence phase, and hence iodide formation is lagged 60 days from primary production [58–61].

Iodide is oxidized to iodate in association with ammonium oxidation in the mixed layer, with the same I:N:C ratio associated with iodide production, and with C and N linked by the Redfield ratio [52]. The partitioning of ammonium oxidation between mixed layer nitrification and nitrification in the deep ocean has been quantified by Yool *et al*. [68], using a global biogeochemical model, and this partitioning is used in the iodine cycling model. Iodide oxidation by this mechanism has a profound effect on the model iodide concentrations, as it results in a spatially variable partial removal of iodide from the mixed layer, and is associated with long

timescales in environments where nitrification is weak/absent and much shorter timescales where nitrification is active. The remaining iodide is subject to removal by ocean mixing and advection and much slower chemical oxidation to iodate. We find that this association of iodide oxidation with mixed layer nitrification gives a much better model fit to observations compared to iodide oxidation over fixed timescales [52].

Perturbation of model parameters and processes shows that primary productivity, mixed layer depth, oxidation, advection, surface fresh water flux and the I:C ratio all have a role in determining surface iodide concentrations. Due to the relatively long residence time of iodide in the mixed layer (months), deep vertical mixing in cold, high latitude Arctic waters results in the dilution of iodide formed by biological activity in the surface ocean, and hence leads to lower surface iodide concentrations (e.g. [61]). Iodide transferred into deeper ocean waters, below a few hundred meters, is oxidized to iodate but the timescales of this oxidation and the return of this water to the surface layer are years to decades. Conversely, in warm, lower latitude waters, greater stratification facilitates the accumulation of higher iodide concentrations.

Figure 4 shows a comparison of the modelled surface iodide field with observations and with the parametrizations of Chance *et al*. [49], MacDonald *et al*. [17] and the machine learning approach of Sherwen *et al*. [51]. The model shows generally good agreement with the observations. It also shows good agreement with the parametrizations in regions where observations exist, but significant differences in the Arctic and subtropical gyres, which are poorly sampled (there are currently no observations in the Arctic). The shallow halocline in the Arctic results in multi-annual residence times for surface waters, allowing iodide to accumulate year-on-year, resulting in high modelled surface concentrations. By contrast, in the highest southern latitudes, stratification is weaker and mixed layer depths are generally deeper, resulting in shorter residence times of surface waters, and more dilution of iodide.

The model predicts quite low iodide in the subtropical gyres, predominantly because of low productivity and therefore slow iodate to iodide conversion, yet relatively rapid nitrification-dependent iodide oxidation. The Chance *et al*. [49] and Sherwen *et al*. [51] parametrizations however predict high iodide in the ocean gyres, consistent with observations at similar latitudes. Advection redistributes iodide within the ocean gyres and supplies iodide to the Arctic. Thus, iodide cannot simply be described by local oceanic conditions, and modelled distributions of iodide are likely to give a more accurate estimate of the ocean surface iodide distribution than methods based on local relationships alone, which may not capture the full range of processes involved. Observations of iodide in currently under-sampled regions, and improved process understanding, are necessary to fully evaluate and develop this prototype iodine cycling model.

Nevertheless, we have used the model to tentatively explore potential future changes in ocean iodide. Specifically, prompted by our bacterial culture experiments which support a link between nitrification and the oxidation of iodide to iodate [67], we have investigated the impact of changes in nitrification rate on sea-surface iodide distribution. Rates and spatial distribution of nitrification in the oceans are influenced by environmental factors such as oxygen level, temperature and pH (see [79]), all of which are currently changing. Some laboratory- and field-based studies indicate that ocean acidification may have a detrimental effect on nitrification, with lower ammonia oxidation rates and slower ammonium-oxidizing bacteria growth rates ([80] and references therein). Beman *et al*. [80] have suggested that ammonia oxidation rates could decline by as much as 3–44% in response to the 0.1 decrease in ocean pH expected over the next 20–30 years. We used our iodine cycling model to investigate the impact of changes of this magnitude by perturbing nitrification rates by +10, −10, −22 and −44%, which in turn altered iodide oxidation rates in the model [67]. We find a global mean sensitivity of 0.13 nM increase in surface iodide for each per cent decrease in nitrification. Figure 5 shows that decreased nitrification rates of the scale predicted by Beman *et al*. [80] could lead to an increase in the concentration of sea-surface iodide across the world's oceans. The largest changes are likely to occur in regions where iodide oxidation is a dominant part of the inorganic iodine cycle such as the subtropical gyres, where they could drive an increase of around 10 nM iodide (equivalent to approx. 10%) [52]. An increase in oceanic iodide will lead to regional-scale decreases in $O_3$ concentrations, through both greater

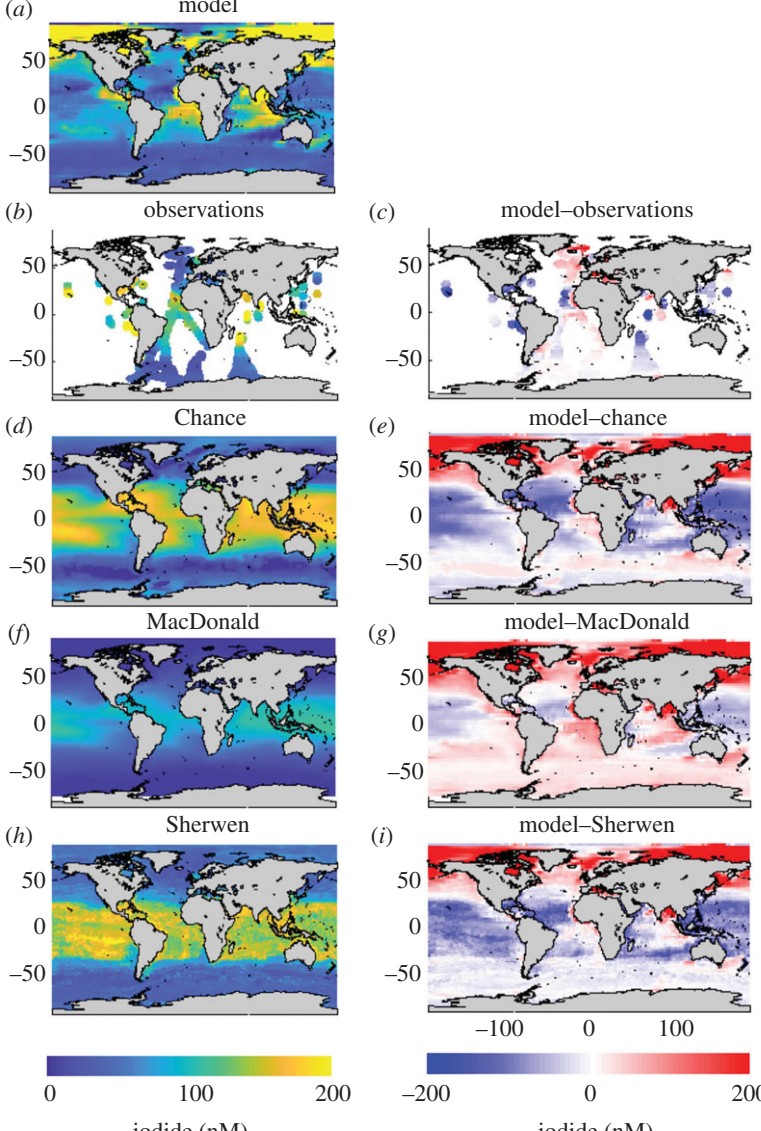

**Figure 4.** Mixed layer iodide concentrations (*a*) predicted from the ocean cycling model [52], (*b*) from observations [49] and from the parametrizations of (*d*) Chance *et al*. [49], (*f*) MacDonald *et al*. [17] and (*h*) Sherwen *et al*. [51]. Differences with the ocean cycling model are shown in the right-hand column. (*a*) Model, (*b*) observations, (*c*) model—observations, (*d*) Chance, (*e*) model—Chance, (*f*) MacDonald, (*g*) model—MacDonald, (*h*) Sherwen and (*i*) model—Sherwen. (Online version in colour.)

$O_3$ deposition to the sea surface and the resulting iodine-initiated catalytic $O_3$-destroying cycles in the atmosphere.

At high latitudes, the dominant iodide loss process is removal from the mixed layer by seasonal mixing, so changes in nitrification rates result in only small, but still significant, changes. In these areas, and elsewhere, changes in ocean mixing and biological productivity in response to climate change are likely to impact iodine speciation. More work is required to examine the impact of acidification and other changing oceanic conditions on iodine speciation over longer time scales.

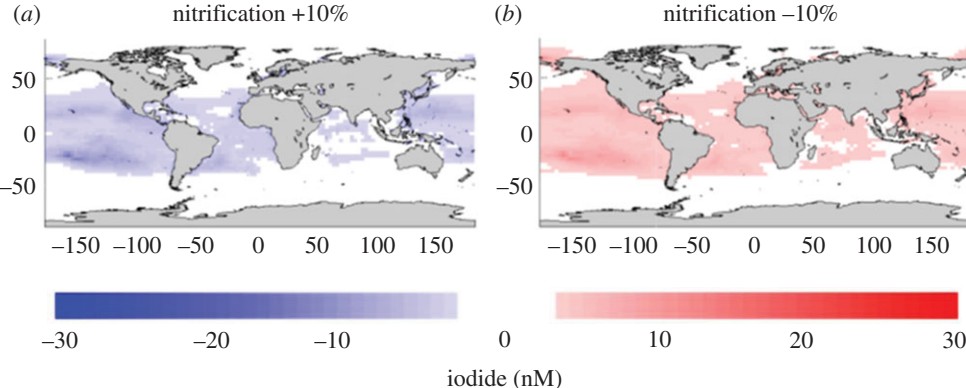

**Figure 5.** Modelled changes in surface $I_{(aq)}^-$ concentration (nM) resulting from (a) +10%, (b) −10%, changes in the rates of nitrification. Negative values on the scale bar indicate a decrease in $I_{(aq)}^-$ concentrations and *vice versa*. From [67]. (Online version in colour.)

# 5. Past and future impacts of iodine on the atmosphere

Given the uncertainties both in the atmospheric chemistry of iodine (e.g. [5]), and in how the iodine precursor emissions may be modified in the real ocean environment compared to the laboratory [29], further studies are required to fully understand iodine emissions and cycling. Despite these uncertainties, the balance of evidence suggests that the release of iodine from the sea surface via the ozone-iodide reaction is the major source of atmospheric iodine. Increasing $O_3$ concentrations since the pre-industrial period (due primarily to increased anthropogenic emissions of nitrogen oxides) imply that atmospheric iodine should be substantially higher now than in the past. Recently, this has been confirmed from records from an Alpine ice core [81] (figure 6) and from a Greenland ice core [82], both showing a tripling in iodine over the latter half of the twentieth century. These results can be broadly explained by increased oceanic iodine emissions from the North Atlantic, and show that iodine's impact on the Northern Hemisphere atmosphere has accelerated over the twentieth century. They also reveal a coupling between anthropogenic pollution and the availability of iodine as an essential nutrient to the terrestrial biosphere. Changes in halogen chemistry have been calculated to reduce by 25% the radiative forcing from increases in ozone since the pre-industrial era, with increased oceanic iodine emissions responsible for about one-third of that [23].

Up until now, while changes in oceanic iodine emissions have been explored in the context of changing anthropogenic surface $O_3$ [83], there have been no attempts to predict how future climate-induced oceanographic changes could impact on surface ocean iodide and hence iodine emissions. Based on the predicted global increases in sea-surface iodide arising from the possible impact of ocean acidification on nitrification, as described in [67], we have estimated the changes in the emission flux using the GEOS-Chem (v. 12.9.1) model.

We have calculated how changes in global sea-surface iodide concentrations scale with the resulting global changes in inorganic oceanic iodine emissions (HOI and $I_2$) using the GEOS-Chem model (v. 12.9.1, [84]), and find that a 1% increase in $[I_{(aq)}^-]$ induces an approximately 0.7% increase in iodine emissions. The scaling is near-linear over environmental concentrations of iodide. These changes are calculated over a short model timescale (3 days), so they give an instantaneous estimate of emissions change without considering any feedback effects of changes to surface ozone concentrations. The results of Hughes *et al*. [67] imply that a change in the average global sea-surface iodide of +5.7% (6.9 nM) could occur if the maximum decline in nitrification proposed by Beman *et al*. [80] over the next few decades takes place. The corresponding global increase in sea–air iodine emissions (HOI and $I_2$) is of the order of 3.6%.

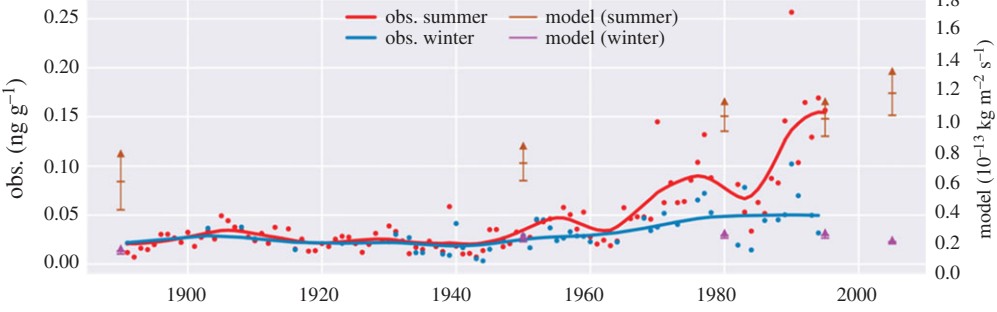

**Figure 6.** Time series of iodine in Col du Dome (CDD) ice core in summer (red) and winter (blue) [81]. Dots are yearly values; solid lines are the first component of a single spectra analysis with a 7-year time window (summer) and robust spline (winter). Brown (summer) and purple (winter) symbols show the modelled mean deposition for 1850, 1950, 1980, 1995 and 2005. The dashes and arrow on the bars show the deposition of inorganic iodine minus HOI (lower dash) and minus half of HOI deposition (middle dash), and the total (top arrow), to reflect uncertainties in post-depositional iodine loss. (Online version in colour.)

Regional increases could be much greater: figure 5 shows an approximately 10% increase in [iodide] in the subtropical gyres, which would result in an increase of iodine emissions of the order of 7%. Exploring this scenario demonstrates how the interaction of global changes such as ocean acidification with the marine iodine cycle has the potential to have impacts on atmospheric chemistry in the relatively short term. Our iodine ocean cycling model indicates that additional factors including primary productivity, biological community structure and vertical mixing all also have a role in determining surface iodide concentrations, so it is anticipated that changes in these processes may cause further changes in atmospheric iodine emissions over the coming decades.

# 6. Conclusion and future developments

Historically, the biogeochemical cycling of iodine has tended to be studied separately, and by different scientific communities, in its marine and atmospheric compartments. The study of iodine on an Earth system scale is extremely challenging because its biogeochemical cycles occur on a vast array of timescales—from seconds for some atmospheric processes to up to millennia in the ocean. While substantial progress has been made in the last decades on developing atmospheric models of iodine cycling, global ocean iodine modelling is in its infancy [52]. There are still large gaps in our basic knowledge that significantly limit how iodine biogeochemistry can be represented. These include the rates and controls of iodine cycling in the ocean including in oxygen-depleted waters, and how iodide present at the very surface of the ocean is quantitatively transformed into iodine emissions to the atmosphere. Major observational gaps which limit our basic understanding include very little laboratory data and no field data on atmospheric HOI, believed to be the major carrier of iodine from the ocean to the atmosphere, and a lack of observations of ocean iodine speciation in some regions, particularly in the subtropical gyres and in the Arctic. The extent of seasonal variation in sea-surface iodide concentrations at any given location is also very poorly constrained.

Although the tools to explore the role of iodine in the Earth system are not yet fully developed, we propose that a consideration of iodine from such a perspective is necessary to understand the linkages and feedbacks between biogeochemical and physical processes in the ocean and ozone (and other oxidants) in the atmosphere, which have policy-relevant impacts arising from emissions of ozone precursors through to climate change, ocean acidification and stratospheric ozone. We have highlighted here the potential impact of ocean acidification on atmospheric iodine emissions in the next two to three decades. Further significant changes could arise over

the coming decades driven by iodine cycling in the ocean through continued ocean acidification, deoxygenation, and reduced productivity, as well as changes in ocean circulation and vertical mixing. These changes in iodine have implications for the management of tropospheric ozone levels by precursor (nitrogen oxides and hydrocarbons) emission control.

Data accessibility. The compiled seawater iodide data from this study are publicly available at BODC (https://doi.org/10.5285/7e77d6b9-83f-41e0-e053-6c86abc069d0) and described in Chance et al. [50].

Authors' contributions. L.J.C. and R.J.C. conceived of the study; L.J.C. coordinated the study and drafted the manuscript; H.H., C.H. and R.J.C. designed and carried out the incubation studies; M.R.W., D.P.S. and T.D.J. participated in the design of the ocean iodine modelling and M.R.W. carried it out; T.S. and M.J.E. designed the atmospheric modelling and T.S. carried it out; T.J.A., L.D.J.H., L.T. and S.M.B. carried out $I_2$ emission experiments; T.S. and R.J.C. synthesized global iodide data; A.M. coordinated cruise participation; T.D.J., R.J.C., M.R.W., A.M. and T.S. critically revised the manuscript.

Competing interests. The authors declare no competing interests.

Funding. This work was funded by the Natural Environment Research Council (NERC), UK, through the grant 'Iodide in the ocean: distribution and impact on iodine flux and ozone loss' (NE/N009983/1 University of York, NE/N01054X/1 University of East Anglia and NE/N009444/1 University of Leicester). L.J.C. also acknowledges funding from the European Research Council (ERC) under the European Union's Horizon 2020 program (Grant agreement no. 833290).

Acknowledgements. Model runs for this work were undertaken on the Viking Cluster, which is a high-performance computer facility provided by the University of York. We are grateful for computational support from the University of York High Performance Computing service, Viking and the Research Computing team. For the ocean model runs we acknowledge the support of resources provided by the High Performance Computing Cluster supported by the Research and Specialist Computing Support service at the University of East Anglia.

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
