## [Peer Review File · Proceedings. Mathematical, Physical, and Engineering Sciences]

Review History

RSPA-2020-0824.R0 (Original submission)

Review form: Referee 1

Is the manuscript an original and important contribution to its field?

Excellent

Is the paper of sufficient general interest?

Excellent

Is the overall quality of the paper suitable?

Excellent

Can the paper be shortened without overall detriment to the main message?

Yes

Do you think some of the material would be more appropriate as an electronic appendix?

No

Do you have any ethical concerns with this paper?

No

Recommendation?

Accept with minor revision (please list in comments)

Comments to the Author(s)

Please see the attached PDF

Review form: Referee 2

Is the manuscript an original and important contribution to its field?

Excellent

Is the paper of sufficient general interest?

Excellent

Is the overall quality of the paper suitable?

Excellent

Can the paper be shortened without overall detriment to the main message?

Yes

Do you think some of the material would be more appropriate as an electronic appendix?

No

Do you have any ethical concerns with this paper?

No

Recommendation?

Accept with minor revision (please list in comments)

Comments to the Author(s)

Carpenter and co-authors present a review of recent findings in surficial iodine chemistry. Such a review is timely and of will be of broad interest to readers in atmospheric sciences, chemical and biological oceanography, and Earth system modeling. The manuscript is well-written, has clear figures, and was a fascinating read. I recommend publication in *Proceedings A*, though would ask the authors to first consider a number of points of clarification, detailed below.

1. Iodine cycling at the ocean-atmosphere interface. I was confused by the description of how iodine and ozone interact to produce IO radicals (Section 1). On lines 63–64 and in Fig. 1, the authors state that the reaction with ozone is a three-step process; iodine compounds are emitted from seawater, are then photolyzed to iodine atoms, and only then react with tropospheric ozone. However, in a number of other places in the manuscript (e.g., lines 137ff., Figure 3), the authors note that tropospheric ozone can invade surface seawater, where it reacts with iodide *in situ*. (This process is alluded to in Fig. 1, but the reactions are not shown.) So I am left wondering, is iodine ‘getting out’ of the ocean, or is ozone ‘getting in’? Is it both? If so, is there any literature describing which pathway is more important? And, the two pathways have different sensitivities that might be important to consider under the future global change scenarios described in Section 5?

2. Ocean iodine cycling model. The authors describe the recent iodine cycling model of Wadley et al. (2020, ref. [51]) as being the first of its kind. However, this statement is completely accurate as there is already a global implementation of the iodine cycle in the cGENIE Earth system model described by Lu et al. (2018; doi:10.1126/science.aar5372). Interestingly, Lu’s model indicates that the iodate [IO_3^-] reduction threshold is $30 \mu\text{mol/kg}$ and that iodide

has a lifetime of ~50 years. It would be worthwhile to compare Lu's model results against the other approaches listed in Table 1, and to discuss the key differences between the parameterizations used by Wadley et al. (2020) and Lu et al. (2018).

3. Deep ocean iodide. In general I felt there was not much discussion of the marine iodine cycle outside of those transformations occurring in the upper ocean. Indeed, the model described by Wadley et al. (2020) seems to prescribe a deep ocean iodide concentration of 0 nmol/kg. However, deep-ocean iodine sources, such as sediments, can release considerable quantities of iodide into the upper mesopelagic (e.g., Kennedy & Elderfield, 1987; doi:10.1016/0016-7037(87)90301-2; Cutter et al., 2018, ref. [66]). Iodide emanating from sedimentary sources may even outcrop at the ocean surface (e.g., in the Bay of Bengal; Chance et al., ref. [46]). Given the relatively long residence time of iodide in the mixed layer, I am curious as to whether this iodide source to the surface is significant – does it rival *in situ* reduction by phytoplankton or is it a mere curiosity?

4. Minor points

- Lines 63: Define rapid – hours, days, months?
- Lines 77–80: Authors state that iodinated organic compounds are not major sources of atmospheric iodine. However, in Fig. 1, these sources are shown to comprise >40 % of the oceanic outward fluxes. This mismatch requires clarification.
- Lines 215–217: This statement needs revising in light of the recent work by Cutter et al. (2018). There are significant accumulations of iodide present below the mixed layer, even relatively far from the continental margin.
- Line 263–266: Our team recently measured iodate reduction in the Eastern Tropical North Pacific OMZ using the radiotracer-incubation method and confirmed that the rates are indeed slow (see Hardisty et al., in press; doi:10.1016/j.epsl.2020.116676).
- Lines 311–313: For comparison with the residence times of iodide in the sea surface, could the authors describe iodine residence times in the troposphere and stratosphere?
- Figure 1: Define $I_{sub}Y_{sub}$ at first use.
- Figure 4: Consider including the model results of Lu et al. (2018) in the comparison.
- Figure 7: This figure is unnecessary; the relationship is linear and already described in words on lines 427–428.

Decision letter (RSPA-2020-0824.R0)

15-Dec-2020

Dear Professor Carpenter,

On behalf of the Editor, I am pleased to inform you that your Manuscript RSPA-2020-0824 entitled "Marine iodine emissions in a changing world" has been accepted for publication subject to minor revisions in Proceedings A. Please find the referees' comments below.

The reviewer(s) have recommended publication, but also suggest some minor revisions to your manuscript. Therefore, I invite you to respond to the reviewer(s)' comments and revise your manuscript.

If possible please submit the revised version of your manuscript within 7 days. If you do not think you will be able to meet this date please let me know in advance of the due date.

To revise your manuscript, log into <https://mc.manuscriptcentral.com/prsa> and enter your Author Centre, where you will find your manuscript title listed under "Manuscripts with

Decisions." Under "Actions," click on "Create a Revision." Your manuscript number has been appended to denote a revision.

You will be unable to make your revisions on the originally submitted version of the manuscript. Instead, revise your manuscript and upload a new version through your Author Centre.

When submitting your revised manuscript, you will be able to respond to the comments made by the referee(s) and upload a file "Response to Referees" in Step 1: "View and Respond to Decision Letter". You can use this to document any changes you make to the original manuscript. In order to expedite the processing of the revised manuscript, please be as specific as possible in your response to the referee(s).

IMPORTANT: Your original files are available to you when you upload your revised manuscript. Please delete any redundant files before completing the submission process.

When uploading your revised files, please make sure that you include the following as we cannot proceed without these:

- 1) A text file of the manuscript (doc, txt, rtf or tex), including the references, tables (including captions) and figure captions. Please remove any tracked changes from the text before submission. PDF files are not an accepted format for the "Main Document".
- 2) A separate electronic file of each figure (tif, eps or print-quality pdf preferred). The format should be produced directly from original creation package, or original software format.
- 3) Electronic Supplementary Material (ESM): all supplementary materials accompanying an accepted article will be treated as in their final form. Note that the Royal Society will not edit or typeset supplementary material and it will be hosted as provided. Please ensure that the supplementary material includes the paper details where possible (authors, article title, journal name). Supplementary files will be published alongside the paper on the journal website and posted on the online figshare repository (<https://figshare.com>). The heading and legend provided for each supplementary file during the submission process will be used to create the figshare page, so please ensure these are accurate and informative so that your files can be found in searches. Files on figshare will be made available approximately one week before the accompanying article so that the supplementary material can be attributed a unique DOI. Alternatively you may upload a zip folder containing all source files for your manuscript as described above with a PDF as your "Main Document". This should be the full paper as it appears when compiled from the individual files supplied in the zip folder.

Article Funder

Please ensure you fill in the Article Funder question on page 2 to ensure the correct data is collected for FundRef (<http://www.crossref.org/fundref/>).

Media summary

Please ensure you include a short non-technical summary (up to 100 words) of the key findings/importance of your paper. This will be used for to promote your work and marketing purposes (e.g. press releases). The summary should be prepared using the following guidelines:

*Write simple English: this is intended for the general public. Please explain any essential technical terms in a short and simple manner.

*Describe (a) the study (b) its key findings and (c) its implications.

*State why this work is newsworthy, be concise and do not overstate (true 'breakthroughs' are a rarity).

*Ensure that you include valid contact details for the lead author (institutional address, email address, telephone number).

Cover images

We welcome submissions of images for possible use on the cover of Proceedings A. Images should be square in dimension and please ensure that you obtain all relevant copyright permissions before submitting the image to us. If you would like to submit an image for consideration please send your image to proceedingsa@royalsociety.org

Once again, thank you for submitting your manuscript to Proceedings A and I look forward to receiving your revision. If you have any questions at all, please do not hesitate to get in touch.

Best wishes
 Raminder Shergill
proceedingsa@royalsociety.org
 Proceedings A

on behalf of
 Professor Liwu Zhang
 Board Member
 Proceedings A

Reviewer(s)' Comments to Author:

Referee: 1

Comments to the Author(s)
 Please see the attached PDF

Referee: 2

Comments to the Author(s)

Carpenter and co-authors present a review of recent findings in surficial iodine chemistry. Such a review is timely and of will be of broad interest to readers in atmospheric sciences, chemical and biological oceanography, and Earth system modeling. The manuscript is well-written, has clear figures, and was a fascinating read. I recommend publication in Proceedings A, though would ask the authors to first consider a number of points of clarification, detailed below.

1. Iodine cycling at the ocean–atmosphere interface.

I was confused by the description of how iodine and ozone interact to produce IO radicals (Section 1). On lines 63–64 and in Fig. 1, the authors state that the reaction with ozone is a three-step process; iodine compounds are emitted from seawater, are then photolyzed to iodine atoms, and only then react with tropospheric ozone. However, in a number of other places in the manuscript (e.g., lines 137ff., Figure 3), the authors note that tropospheric ozone can invade surface seawater, where it reacts with iodide *in situ*. (This process is alluded to in Fig. 1, but the reactions are not shown.) So I am left wondering, is iodine ‘getting out’ of the ocean, or is ozone ‘getting in?’ Is it both? If so, is there any literature describing which pathway is more important? And, the two pathways have different sensitivities that might be important to consider under the future global change scenarios described in Section 5?

2. Ocean iodine cycling model. The authors describe the recent iodine cycling model of Wadley et al. (2020, ref. [51]) as being the first of its kind. However, this statement is completely accurate as there is already a global implementation of the iodine cycle in the cGENIE Earth system model described by Lu et al. (2018; doi:10.1126/science.aar5372). Interestingly, Lu’s model

indicates that the iodate [O^{2-}] reduction threshold is $30 \mu\text{mol/kg}$ and that iodide has a lifetime of ~ 50 years. It would be worthwhile to compare Lu's model results against the other approaches listed in Table 1, and to discuss the key differences between the parameterizations used by Wadley et al. (2020) and Lu et al. (2018).

3. Deep ocean iodide. In general I felt there was not much discussion of the marine iodine cycle outside of those transformations occurring in the upper ocean. Indeed, the model described by Wadley et al. (2020) seems to prescribe a deep ocean iodide concentration of 0 nmol/kg . However, deep-ocean iodine sources, such as sediments, can release considerable quantities of iodide into the upper mesopelagic (e.g., Kennedy & Elderfield, 1987; doi:10.1016/0016-7037(87)90301-2; Cutter et al., 2018, ref. [66]). Iodide emanating from sedimentary sources may even outcrop at the ocean surface (e.g., in the Bay of Bengal; Chance et al., ref. [46]). Given the relatively long residence time of iodide in the mixed layer, I am curious as to whether this iodide source to the surface is significant – does it rival *in situ* reduction by phytoplankton or is it a mere curiosity?

4. Minor points

- Lines 63: Define rapid – hours, days, months?
- Lines 77–80: Authors state that iodinated organic compounds are not major sources of atmospheric iodine. However, in Fig. 1, these sources are shown to comprise $>40\%$ of the oceanic outward fluxes. This mismatch requires clarification.
- Lines 215–217: This statement needs revising in light of the recent work by Cutter et al. (2018). There are significant accumulations of iodide present below the mixed layer, even relatively far from the continental margin.
- Line 263–266: Our team recently measured iodate reduction in the Eastern Tropical North Pacific OMZ using the radiotracer-incubation method and confirmed that the rates are indeed slow (see Hardisty et al., in press; doi:10.1016/j.epsl.2020.116676).
- Lines 311–313: For comparison with the residence times of iodide in the sea surface, could the authors describe iodine residence times in the troposphere and stratosphere?
- Figure 1: Define $\langle Y \rangle$ at first use.
- Figure 4: Consider including the model results of Lu et al. (2018) in the comparison.
- Figure 7: This figure is unnecessary; the relationship is linear and already described in words on lines 427–428.

Decision letter (RSPA-2020-0824.R1)

28-Jan-2021

Dear Professor Carpenter

I am pleased to inform you that your manuscript entitled "Marine iodine emissions in a changing world" has been accepted in its final form for publication in Proceedings A.

Our Production Office will be in contact with you in due course. You can expect to receive a proof of your article soon. Please contact the office to let us know if you are likely to be away from e-

mail in the near future. If you do not notify us and comments are not received within 5 days of sending the proof, we may publish the paper as it stands.

Open access

You are invited to opt for open access, our author pays publishing model. Payment of open access fees will enable your article to be made freely available via the Royal Society website as soon as it is ready for publication. For more information about open access please visit <https://royalsociety.org/journals/authors/which-journal/open-access/>. The open access fee for this journal is £1700/\$2380/€2040 per article. VAT will be charged where applicable.

Note that if you have opted for open access then payment will be required before the article is published – payment instructions will follow shortly.

If you wish to opt for open access then please inform the editorial office (proceedingsa@royalsociety.org) as soon as possible.

Under the terms of our licence to publish you may post the author generated postprint (ie. your accepted version not the final typeset version) of your manuscript at any time and this can be made freely available. Postprints can be deposited on a personal or institutional website, or a recognised server/repository. Please note however, that the reporting of postprints is subject to a media embargo, and that the status the manuscript should be made clear. Upon publication of the definitive version on the publisher's site, full details and a link should be added.

You can cite the article in advance of publication using its DOI. The DOI will take the form: 10.1098/rspa.XXXX.YYYY, where XXXX and YYYY are the last 8 digits of your manuscript number (eg. if your manuscript number is RSPA-2017-1234 the DOI would be 10.1098/rspa.2017.1234).

For tips on promoting your accepted paper see our blog post: <https://royalsociety.org/blog/2020/07/promoting-your-latest-paper-and-tracking-your-results/>

On behalf of the Editor of Proceedings A, we look forward to your continued contributions to the Journal.

Sincerely,
Raminder Shergill
proceedingsa@royalsociety.org